# MFSFNet: Multi-Scale Feature Subtraction Fusion Network for Remote Sensing Image Change Detection

**Zhiqi Huang** [1,2,3] **and Hongjian You** [1,2,*]

1 Aerospace Information Research Institute, Chinese Academy of Sciences, Beijing 100094, China; huangzhiqi21@mails.ucas.ac.cn

2 Key Laboratory of Technology in Geo-Spatial Information Processing and Application System, Chinese Academy of Sciences, Beijing 100190, China

3 School of Electronic, Electrical and Communication Engineering, University of Chinese Academy of Sciences, Beijing 101408, China

\* Correspondence: hjyou@mail.ie.ac.cn

**Abstract:** Change detection plays a crucial role in remote sensing by identifying surface modifications between two sets of temporal remote sensing images. Recent advancements in deep learning techniques have yielded significant achievements in this field. However, there are still some challenges: (1) Existing change feature fusion methods often introduce redundant information. (2) The complexity of network structures leads to a large number of parameters and difficulties in model training. To overcome these challenges, this paper proposes a Multi-Scale Feature Subtraction Fusion Network (MFSF-Net). It comprises two primary modules: the Multi-scale Feature Subtraction Fusion (MFSF) module and the Feature Deep Supervision (FDS) module. MFSF enhances change features and reduces redundant pseudo-change features. FDS provides additional supervision on different scales of change features in the decoder, improving the training efficiency performance of the network. Additionally, to address the problem of imbalanced samples, the Dice loss strategy is introduced as a means to mitigate this issue. Through comprehensive experiments, MFSF-Net achieves an F1 score of 91.15% and 95.64% on LEVIR-CD and CDD benchmark datasets, respectively, outperforming six state-of-the-art algorithms. Moreover, it attains an improved balance between model complexity and performance, showcasing the efficacy of the proposed approach.

**Keywords:** subtraction fusion; change detection; remote sensing images; multi-scale features

## 1. Introduction

The process of change detection in remote sensing images involves the detection of modifications that occur on the Earth's surface between two separate sets of temporal remote sensing images. This task holds significant importance in remote sensing [1]. By enabling effective monitoring of surface changes, the application of remote sensing image change detection spans various fields and domains. such as urban built-up area expansion monitoring [2], natural disaster assessment [3], and environmental monitoring [4].

Based on the analysis unit, conventional approaches to remote sensing image change detection can be classified into two categories: pixel-based and object-based methods. Pixel-based methods operate on a pixel-by-pixel basis, extracting spectral and texture features from the input images pixel by pixel. They subsequently utilize predefined thresholds to identify change areas for each pixel. Pixel-based methods include arithmetic image differencing [5], change vector analysis based on transformations [6,7], principal component analysis [8,9], and independent component analysis [10]. Object-based change detection methods first segment the images into super-pixel objects using image segmentation techniques based on spectral and texture features (segmentation methods such as quadtree-based segmentation [11], multi-resolution segmentation [12], etc.). Then, in order to derive the outcomes of change detection, the segmented results from different time

periods are compared in object-based methods. In contrast to the pixel-based approach, object-based methods consider contextual information but are more affected by the results of image segmentation. Additionally, both pixel-based and object-based approaches need significant manual intervention and are prone to pseudo-changes caused by sensor and illumination conditions.

Over the past few years, deep learning methods have received considerable recognition within the remote sensing domain, prompting researchers to integrate these techniques into various tasks related to remote sensing, including scene classification [13,14], semantic segmentation [15–17], object detection [18,19], and change detection [20–24], achieving remarkable performance. Deep learning-based approaches for change detection tasks have the ability to incorporate spatial contextual information during pixel-level change identification in the images. CNN (Convolutional neural network) models, with their excellent feature representation capability and end-to-end simplicity, not only reduce manual intervention but also enhance the accuracy and generalization. CNN-based change detection methods transform the two temporal remote sensing images into high-level features, extract semantic context of change regions by fusing the features of the two time-phased images, and mitigate artificial errors stemming from preprocessing. There are two types of CNN-based change detection methods, categorized based on the fusion strategy utilized: image-level fusion [25–29] and feature-level fusion [30–37]. In image-level fusion networks, the two remote sensing images from different temporal input as a whole into CNN to obtain a representation of image differences. However, the absence of deep feature extraction from individual temporal images can result in boundary disturbances within the predicted outcomes, thereby constraining the accuracy of change detection. In contrast to image-level fusion networks, feature-level fusion networks employ two networks with shared parameters. These networks independently learn features from individual temporal images and then combine these features as inputs to the classifier, overcoming the limitations mentioned above.

With the development of satellites and airborne sensors, more detailed and objective representations of the surface can be observed, thus offering finer-grained data for detecting surface changes. However, the diversity of surface features, especially the variability in object shapes, the complexity of background objects, as well as differences in weather conditions, imaging angles, and sensors, can easily lead to false detections or missed detections of actual change areas. Additionally, objects in remote sensing images exhibit different sizes, and a robust and generalizable model should be capable of handling various object scales. Moreover, change detection tasks often encounter a substantial class imbalance problem, where the count of unchanged pixels significantly outweighs the count of changed pixels. This presents a significant challenge in deep learning-based change detection methods for high-resolution remote sensing, as it becomes crucial to effectively extract and utilize the abundant feature information from high-resolution imagery to mitigate the influence of pseudo-changes and enhance the accuracy of detection.

To address this challenge and achieve better representation capabilities of deep features, designing deeper and more complex feature extraction networks has gotten significant attention as a primary research focus. Many researchers have put forward several enhanced models to achieve more discriminative feature representations, such as combining Generative Adversarial Networks (GAN) [38–40] or Recurrent Neural Networks (RNN) [41,42], or using feature extraction models based on the Transformer architecture [43–45] to expand the receptive field. Some studies focus on the effective utilization of features, such as using spatial or channel attention mechanisms [30–32,36,46] or employing multi-scale feature fusion for feature enhancement [25,29,47–49]. However, along with the increased complexity of the models, there is a proliferation of parameters and redundant feature information. This not only imposes a heavy burden on model training but also increases the risk of pseudo-changes detections due to the presence of excessive redundant information.

In light of the aforementioned challenges, we present a novel approach called the Multi-Scale Feature Subtraction Fusion Network (MFSFNet). Our network is constructed based on the following four requirements. Firstly, the model should maximize the utilization of feature information derived from the dual-temporal imagery, emphasizing real change areas while reducing the generation of redundant information to minimize false detections caused by complex backgrounds or imaging differences. Secondly, the network must be able to effectively represent features of diverse structures and different sizes of objects. Thirdly, the model should be easy to train, avoiding the issue of gradient vanishing. Lastly, it is essential for the model to effectively tackle the issue of sample imbalance and enhance the accuracy of detection. To address the first two requirements, we extract the dual-temporal imagery multi-scale features and design the Multi-scale Feature Subtraction Fusion (MFSF) module to fuse these features. Unlike existing fusion strategies, our subtraction fusion strategy enhances change features while reducing the interference of redundant features. To meet the third requirement, we use ConvnNext v2-Atto, as the lightweight multi-scale feature encoder with less parameters. We also introduce the Feature Deep Supervision (FDS) module in the decoder to provide additional supervision for deep change features, which improves the model's feature extraction capability while accelerating convergence. To address the fourth requirement, we incorporate Dice loss into the loss function to mitigate the imbalance between change and non-change pixels. With these efforts, our network can effectively capture change features in high-resolution imagery.

The primary contributions of this work can be summarized as follows:

1.  We propose the MFSFNet for high-resolution remote sensing image change detection. This network enhances change features and reduces redundant pseudo-change features through a multi-scale subtraction fusion strategy.
2.  We utilize a lightweight feature extraction network and introduce a novel deep supervision strategy in the change decoder, which enhances the training performance of the network.

The paper is structured as follows: Section 1 provides an introduction to the background and problem addressed in this study. Section 2 discusses relevant literature and related works. Section 3 presents the comprehensive details of MFSFNet. Section 4 describes the experimental design, parameter settings, and analysis of the obtained results. Section 5 offers a discussion of our method. Finally, Section 6 concludes the paper, summarizing the key findings and contributions.

## 2. Related Works

### 2.1. Encoder in Change Detection Task

To enhance the learning and representing change features, recent studies have concentrated on improving the architecture and capabilities of change detection encoders. One approach is to increase the depth of the encoder by incorporating more convolutional layers. Regarding network structures, Peng et al. introduced the Unet++_MSOF [25], which builds upon the Unet++ architecture [50,51]. Unet++_MSOF employs a technique of combining two individual temporal images, each having C channels, to create a 2C data representation. This combined data representation is then inputted into the network for the purpose of identifying the changed regions. This method allows for the incorporation of temporal information and enhances ability to capture changes between the two images. Zhan et al. [52] introduced the use of Siamese convolutional networks for change detection tasks, which are composed of twin networks that share weights and are trained to identify similarities or differences between two input samples. The concept of end-to-end training for change detection was introduced by Tao et al. [37]. In their work, FC-EF employed a fully connected layer to establish a connection between the two temporal images, serving as input to the network. On the other hand, FC-Siam-conc and FC-Siam-diff employed a Siamese structure, enabling direct processing of the temporal images. This approach facilitated the integration of temporal information and resulted in improved change detection performance. Chen and Shi [53], as well as Chen et al. [31], introduced STANet

and DASNet, respectively, using ResNet [54] as the backbone. These architectures took advantage of ResNets powerful feature extraction capabilities to boost the effectiveness of change detection models. To address the inherent localness of convolutional operations, Zhang et al. [55] adopted dilated convolutions instead of traditional convolutions and achieved promising results. Furthermore, dense connections between features have been demonstrated to improve the performance of networks. Thus, Fang et al. [29] and Peng et al. [46] employed dense connections between features at layers, enhancing the network's capabilities. Additionally, for the task of change detection in multispectral time-series images, 3D CNN models are developed to handle spatial-spectral-temporal features [56]. However, it is important to acknowledge that these methods introduce additional complexity and computational costs to change detection algorithms. Therefore, it becomes crucial to carefully consider and strike a balance between achieving high detection accuracy and managing the computational complexity involved.

### 2.2. Multi-Scale Feature Fusion in Change Detection Task

The incorporation of multi-scale feature fusion modules has gained significant attention to effectively represent the multi-scale characteristics of geo-objects. One notable approach is PSPNet-CONC [49], which introduced the PSP module [17] for multi-scale feature extraction. The PSP module allows the network to capture contextual information at multiple scales, enabling a more comprehensive representation of the objects in the image. Another approach is utilized by Unet++_MSOF [25] and NestNet [48]. This module allows for the integration of features from various levels of the network, combining both detailed and high-level information to enhance the precision of change detection. To further enhance the detection accuracy, these methods also employ a multi-output fusion strategy. For instance, ADS-Net [47] integrates features from different branches in the decoder and calculates the F1 score for each scale output. Subsequently, the change maps are weighted according to the F1 score to emphasize more reliable detections. It is important to mention that the fusion strategies mentioned above are based on addition, although commonly used, may introduce noise in the fused results and increase the complexity of the model. This can make convergence challenging and potentially lead to false change detections. Therefore, researchers need to carefully balance the complexity and fusion strategies to ensure accurate and reliable results.

Table 1 demonstrates our summary and comparison analysis of encoder and feature fusion strategies for above change detection methods.

**Table 1.** The literature review summary and comparison analysis.

| Method | Encoder Architecture | Feature Fusion Strategy | Proposed Year |
| --- | --- | --- | --- |
| Siamese convolutional networks [52] | Twin networks with shared weights. | N/A | 2017 |
| FC-EF, FC-Siam-conc, FC-Siam-diff [37] | Fully connected layer skip connecting. | Concatenation | 2018 |
| Unet++_MSOF [25] | Multi-scale feature for combining low-level and high-level information. | Addition | 2019 |
| Dilated convolutions [55] | Twin networks with shared weights and dilated convolutions used instead of traditional convolutions. | Concatenation | 2019 |
| STANet [53] | Twin networks with shared weights and ResNet used as the backbone. | Addition | 2020 |
| PSPNet-CONC [49] | Introduces PSP module for multi-scale feature extraction and contextual information capture. | Addition | 2020 |

**Table 1.** *Cont.*

| Method | Encoder Architecture | Feature Fusion Strategy | Proposed Year |
|---|---|---|---|
| DASNet [31] | Twin networks with shared weights and ResNet used as the backbone. | Addition | 2021 |
| NestNet [48] | Multi-scale feature for combining low-level and high-level information. | Addition | 2021 |
| ADS-Net [47] | Multi-scale feature for combining low-level and high-level information. | Addition | 2021 |
| SNUNet-CD [29] | Employs dense connections between features at layers. | Addition | 2022 |
| 3D CNN [56] | A 3D CNN based on a pretrained 2D CNN | Concatenation | 2022 |

## 3. Methods

This section presents a comprehensive overview of the MFSFNet (Multi-Scale Feature Subtraction Fusion Network) architecture, highlighting its key components and functionality. Firstly, we present the MFSFNets flowchart and overall architecture (refer to Figures 1 and 2). Then, we describe in detail the MSSF module and the Feature Deep Supervision (FDS) module that we have designed. Lastly, we define the loss function.

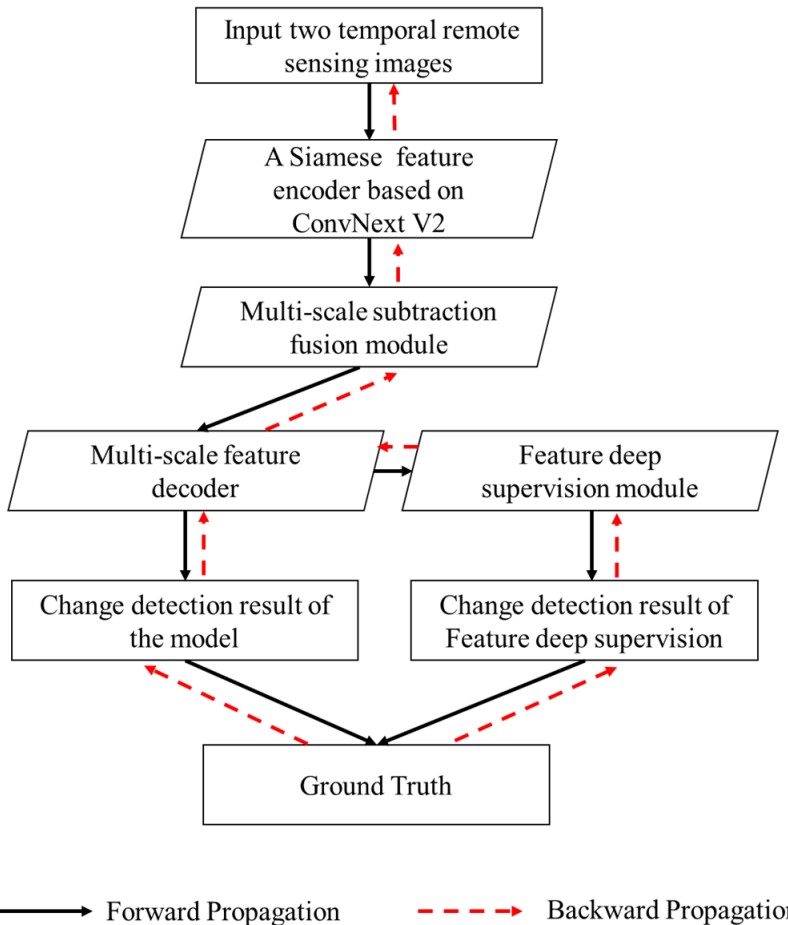

**Figure 1.** The Flowchart of MFSFNet.

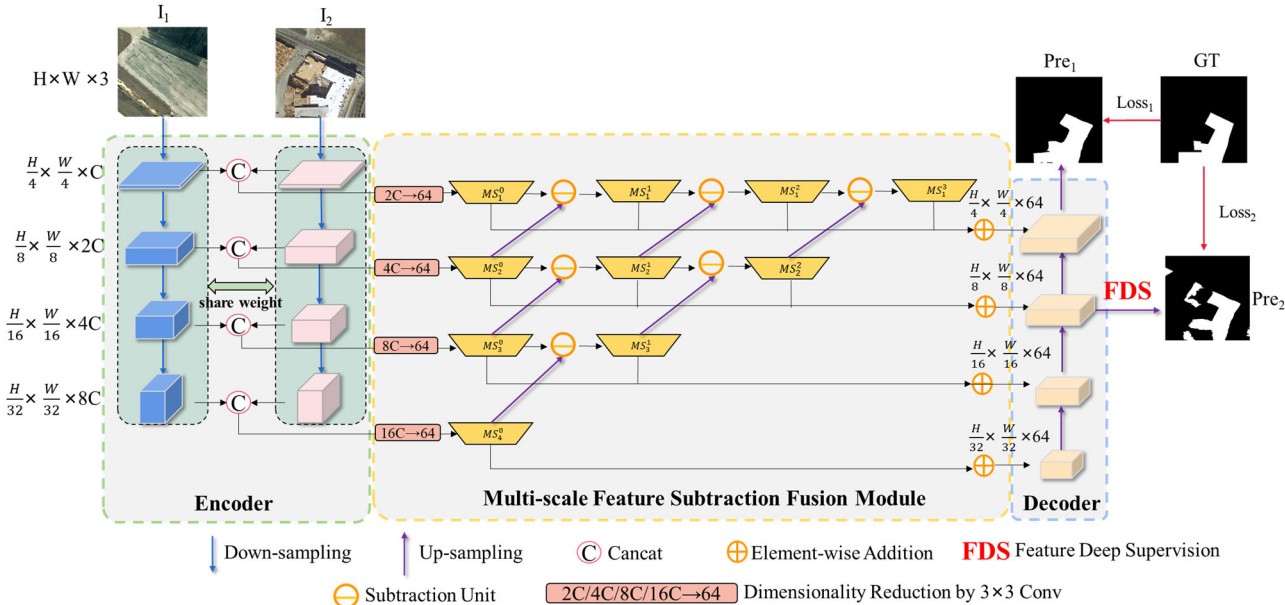

**Figure 2.** The overall architecture of MFSFNet. $I_1$ and $I_2$ are two input remote sensing images of different time periods. H is the height of the input image, W is the width of the image, C is the number of channels. $MS_j^i$ represents feature fusion result, where $j = 1, 2, 3, 4$ indices the scale level, and $i = 0, 1, \ldots, 4 - j$ indices the number of feature fusion operations at each scale level. $Pre_1$ denotes the final change detection result. $Pre_2$ denotes the change detection result of the feature deep supervision. GT denotes the ground truth of change detection result. $Loss_1$ and $Loss_2$ denote the loss between the final change detection result and the true value of the model, and the loss between the change detection result and the true value of the deep supervised change detection result, respectively.

### 3.1. Flowchart and Overall Architecture of MFSFNet

As the flowchart of MFSFNet shown in Figure 1, firstly, the inputs of MFSFNet are two temporal remote sensing images. Secondly, the images are fed into a weight-shared Siamese CNN to extract multi-scale features based on ConvNext V2. Thirdly, multi-scale features are fused by multi-scale subtraction fusion module. Fourthly, the fused features are decoded by a multi-scale feature decoder. The multiscale feature decoder has two branches, one that outputs the change detection results of the model, and a feature deep supervision branch that generates an additional change detection result through feature deep supervision. Lastly, both results are used with ground truth to compute the loss and perform loss back propagation. The module of feature deep supervision subjects the model to additional supervision. The black solid arrows in the figure show the sinusoidal propagation path of the data and the red dashed arrows show the backward propagation path of the gradient.

Figure 2 illustrates the overall architecture of MFSFNet. MFSFNet is composed of three components: the Siamese encoder, the Multi-scale Subtraction Feature Fusion (MSSF), and the decoder. The decoder includes the Feature Deep Supervision (FDS) module for multi-scale supervision. Firstly, we employ a ConvNext architecture [57,58] as the feature encoder to obtain multi-level features while enhancing the generalization of feature representations, enabling the model to tackle the challenges posed by diverse land cover structures. Secondly, the MSSF module performs feature fusion on different scales. It reduces feature redundancy while emphasizing different-sized objects and minimizing interference from complex background information. This helps to generate more accurate boundaries. Next, the MSSF module's output features are bottom-up integrated by the decoder as part of the feature decoding process to produce change features. Subsequently, FDS is applied to predict change detection results on different scales of the change features. Multiple predictions are compared with the label, and losses are computed accordingly.

Additionally, we incorporate the Dice Loss to increase the model's attention to object boundaries and alleviate issues related to irregular or misaligned boundaries. Finally, the model achieves convergence through loss backpropagation, resulting in improved change detection results. Overall, the MFSFNet network leverages the Siamese feature encoder, MSSF module, and decoder module to effectively fuse multi-scale features and obtain accurate results. The incorporation of the FDS module enhances the training process by providing multi-scale supervision. The addition of the Dice Loss promotes better handling of object boundaries. Through loss propagation, the model achieves convergence and produces superior change detection outcomes.

### 3.2. The Siamese Feature Encoder

The Siamese feature encoder is responsible for extracting multi-scale change features and semantic features that are essential for change detection. To achieve a generalized representation of both semantic and change features, the improved version of the ConvNext network, called ConvNext V2-Atto [58], is employed in MFSFNet. ConvNext V2-Atto is a lightweight variant of ConvNextV2, with a small number of network parameters while maintaining good feature generalization. Compared to ConvNextV1 [57], ConvNext V2 incorporates a fully convolutional masked autoencoder (FCMAE) framework similar to MAE [59] for pre-training, which enhances the feature extraction capability. The feature encoder based on ConvNext V2-Atto is illustrated in Figure 3a.

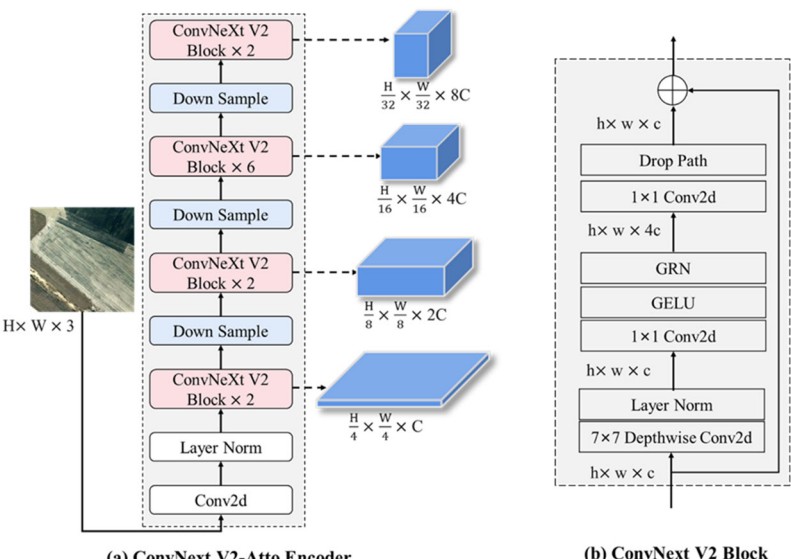

**Figure 3.** The Details of the Encoder.

The basic unit of the ConvNext V2-Atto encoder is the ConvNext V2 Block. As shown in Figure 3b, within each ConvNext V2 Block, the input feature map $F_0$ of size $h \times w \times c$ undergoes a $7 \times 7$ Depthwise convolution [60] and a Layer Norm layer [61], resulting in the feature map $F_1$ of size $h \times w \times c$. Then, a $1 \times 1$ convolution layer, GeLU activation layer [62], and Global Response Normalization (GRN) layer [58] are applied to get the feature map $F_2$ of size $h \times w \times 4c$. The channel count is then restored to $c$ using a $1 \times 1$ convolution layer and Drop Path layers are employed to prevent overfitting to get $F_3$. Finally, an element-wise addition operation is performed between $F_3$ and the input $F_0$, resulting in a feature map $F$ of size $h \times w \times c$. In the ConvNext V2-Atto encoder, four groups of ConvNext V2 Blocks with a ratio of {2:2:6:2} are combined with down-sampling layers to generate four feature maps at different scales: size of $\frac{H}{4} \times \frac{W}{4} \times C$, $\frac{H}{8} \times \frac{W}{8} \times 2C$, $\frac{H}{16} \times \frac{W}{16} \times 4C$, $\frac{H}{32} \times \frac{W}{32} \times 8C$, respectively.

For MFSFNet, two shared-weight ConvNext V2-Atto Encoders are used to get multi-scale features for the two images. The images from two time periods, denoted as $H \times W \times C$ image $I_1$ and $I_2$, are inputted into the twin ConvNext V2-Atto Encoders. This allows for obtaining semantic features for each image at four different scales. Specifically, for $I_1$, the

multi-scale semantic features are $F_1^1$ of size $\frac{H}{4} \times \frac{W}{4} \times C$, $F_2^1$ of size $\frac{H}{8} \times \frac{W}{8} \times 2C$, $F_3^1$ of size $\frac{H}{16} \times \frac{W}{16} \times 4C$, and $F_4^1$ of size $\frac{H}{32} \times \frac{W}{32} \times 8C$. Similarly, $I_2$ yields the multi-scale semantic features $F_1^2, F_2^2, F_3^2$, and $F_4^2$, which have the same size as the corresponding features of $I_1$.

To obtain multi-scale change features for the objects, the semantic features from the two images are concatenated along the channel dimension at each scale. This concatenation ensures that the change features have the same width and height as the semantic features, but with the channel count being twice that of the semantic features of a single image. Finally, we obtain change features $F_1^{Change}$, $F_2^{Change}$, $F_3^{Change}$ and $F_4^{Change}$ of sizes $\frac{H}{4} \times \frac{W}{4} \times 2C$, $\frac{H}{8} \times \frac{W}{8} \times 4C$, $\frac{H}{16} \times \frac{W}{16} \times 8C$, and $\frac{H}{32} \times \frac{W}{32} \times 16C$, respectively. These change features serve as the input to the multi-scale subtraction feature fusion module.

### 3.3. Multi-Scale Feature Subtraction Fusion (MSSF) Module

To enhance the model's generalization ability for objects at various scales and prioritize features relevant to changes while suppressing irrelevant ones, we introduce the Multi-scale Feature Subtraction Fusion (MFSF) module. This module combines the multi-scale features from the encoder and employs a subtraction fusion method. By subtracting corresponding features from different scales, redundant information during feature fusion is reduced. This subtraction fusion approach enhances the representation of relevant features associated with changes and helps mitigate the interference caused by complex backgrounds. The MFSF module emphasizes discriminative information and improving the accuracy.

To facilitate the fusion of different-scale features, we first employ a $3 \times 3$ convolutional layer to lessen the dimension of the multi-scale change features $F_1^{Change}$, $F_2^{Change}$, $F_3^{Change}$, and $F_4^{Change}$ to 64, resulting in $MS_1^0$, $MS_2^0, MS_3^0$, and $MS_4^0$. Then, we apply subtraction units to fuse the features not only within the same scale but also between adjacent scales. The subtraction units enhance the change features and emphasize the change regions at different scales.

The subtraction unit (Figure 4) takes inputs $F_A$ and $F_B$. $F_A$ is derived from the features $MS_j^{i-1}$ at the same scale, while $F_B$ is obtained by up-sampling the features $MS_{j+1}^{i-1}$ from the adjacent scale to match the size of $F_A$. The feature fusion is achieved by Equation (1):

$$MS_j^i = SU(F_A, F_B) = Conv\left(abs\left(MS_j^{i-1} \ominus Upsample\left(MS_{j+1}^{i-1}\right)\right)\right) \tag{1}$$

where $j = 1, 2, 3, 4$ indices the scale level, and $i = 0, 1, \ldots, 4 - j$ indices the number of feature fusion operations at each scale level. $Conv(\cdot)$ denotes a $3 \times 3$ convolution, $abs(\cdot)$ represents the absolute value operation, $\ominus$ indicates element-wise subtraction, and $Upsample(\cdot)$ denotes the up-sampling layer. $MS_j^i$ represents feature fusion result, which has the same size as $MS_j^{i-1}$.

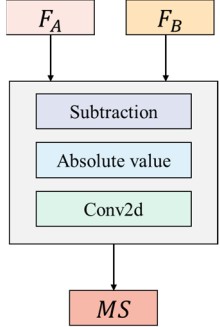

**Figure 4.** Subtraction Unit.

The absolute value operation in the subtraction unit serves a similar purpose as an activation function. When performing fusion operations between features of different scales, the features represented by operation results with the same absolute values indicate

the same feature differences. Therefore, the absolute value operation is used for activation. If the commonly used ReLU activation function is employed, the negative values would be discarded directly, overlooking a portion of the feature differences, which would affect the effectiveness of the subtraction fusion.

After the multi-scale feature fusion, at each scale $j$, the features $MS_j^i$ are aggregated using element-wise addition to further enhance the focus on the changed features. The computation is performed according to Equation (2):

$$SF_j = \sum_{i=0}^{4-j} MS_j^i \tag{2}$$

where $j = 1, 2, 3, 4$ indices the scale level, $MS_j^i$ represents the feature fusion result after i iterations of subtraction units at scale $j$. $SF_j$ denotes the result of adding features at scale j, and its size is the same as $MS_j^i$. $SF_j$ will serve as the input to the decoder module.

*3.4. Decoder and Feature Deep Supervision Module*

Due to the varying expressive capabilities of different feature levels in change detection tasks, we employ a FDS module in the multi-scale decoder. By introducing additional supervisory signals at different levels of the network, the network is able to better utilize multi-level feature representations during the learning process.

The decoder of MFSFNet is composed of four stages and a FDS module (Figure 5). Each stage includes a $3 \times 3$ convolutional, a batch normalization (BN), a ReLU activation, and an up-sampling. Stages 1 to 3 progressively up-sample the channel dimensions of the input feature maps by a factor of 2, while maintaining a consistent channel dimension of 64. This is done to enable element-wise addition with the multi-scale features from MSSF. In Stage 4, the fused feature map of size $\frac{H}{4} \times \frac{W}{4} \times 64$ is converted to a change detection result $Pre_1$ of size $H \times W \times 1$, which serves as one of the decoder's outputs. It is then compared with the ground truth (GT) labels to calculate $Loss_1$.

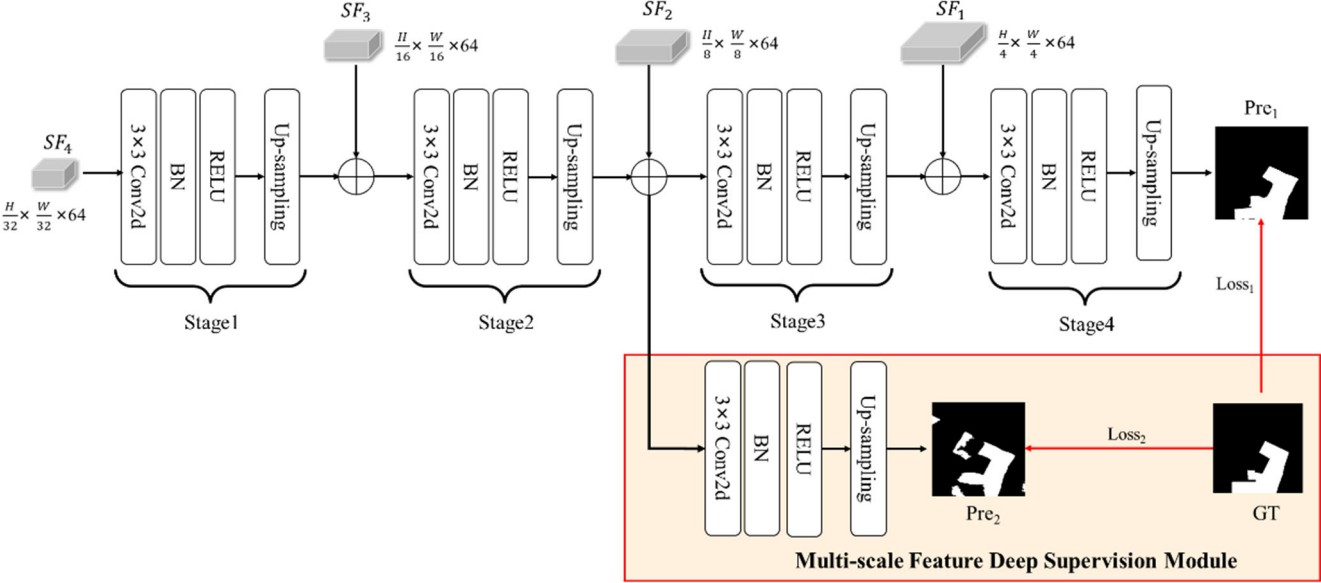

**Figure 5.** Decoder and Feature Deep Supervision Module. $SF_j$ denotes the result of adding features at scale j, where $j = 1, 2, 3, 4$ indices the scale level. H and W denote the length and width of the feature map. $Pre_1$ denotes the final change detection result. $Pre_2$ denotes the change detection result of the feature deep supervision. GT denotes the ground truth of change detection result. $Loss_1$ and $Loss_2$ denote the loss between the final change detection result and the true value of the model, and the loss between the change detection result and the true value of the deep supervised change detection result, respectively.

Additionally, when implementing deep supervision, we took into consideration that the feature map sizes of Stage 3 and Stage 4 are closer to the size of the original image compared to Stage 1 and Stage 2. The larger feature maps indicate the presence of more feature information. Therefore, in addition to supervising the output of Stage 4, supervising Stage 3 may yield better results than supervising Stage 1 and Stage 2. This idea is supported by the results of the ablation experiments in Section 5.1. As a result, we apply deep supervision to the input feature map of Stage 3. In FDS module, Stage 2's output is fused with $SF_2$ and passed through a series of a $3 \times 3$ convolutional, a BN, a ReLU activation, and an up-sampling. This generates a change detection result $Pre_2$ of size H $\times$ W $\times$ 1, which is compared with the labels to calculate $Loss_2$. This process represents the multi-scale feature deep supervision. Finally, after training, the decoder of MFSFNet uses the $Pre_1$ result as the model's change detection output.

### 3.5. Loss Function of MFSFNet

The loss function of MFSFNet consists of two parts: the difference between the decoder's final output $Pre_1$ and the GT, denoted as $Loss_1$, and the difference between the output $Pre_2$ of the multi-scale deep supervision module and GT, denoted as $Loss_2$. The calculation of both losses follows the same approach.

In MFSFNet, the proposed approach incorporates the Dice loss [63] along with the binary cross-entropy (BCE) loss to mitigate the issue of class imbalance and alleviate the boundary fuzziness in the results. Therefore, the calculation of $Loss_1$ and $Loss_2$ is described by Equation (3).

$$Loss_i(\alpha, \beta) = \alpha L_{BCE}(Pre_i, GT) + \beta L_{dice}(Pre_i, GT), \tag{3}$$

where $Pre_i$ $(i = 1, 2)$, represents the binary maps of the two change detection predictions. $Loss_i$ $(i = 1, 2)$, represents the losses of the two predictions. $L_{BCE}$ denotes the binary cross-entropy loss. $L_{dice}$ refers to the Dice loss, calculated as described in Equation (5). The coefficients $\alpha$ and $\beta$ are used, with the common setting in this paper being $\alpha = 0.6$ and $\beta = 0.4$.

$L_{BCE}$ can be calculated as follows:

$$L_{BCE} = -\frac{1}{N} \sum_{i,j=0}^{N} \left( y_{i,j} \log x_{i,j} + (1 - y_{i,j}) \log(1 - x_{i,j}) \right), \tag{4}$$

where $y_{i,j}$ denotes the ground truth label and $x_{i,j}$ denotes the predicted probability of change at a specific point $(i,j)$.

The Dice loss [63] is a metric that can calculated the similarity between two sets from a global viewpoint. It is particularly useful in scenarios where there are only a few positive samples in the image, as it remains effective in such cases. By utilizing the Dice loss, the issue of blurred boundaries caused by the class imbalance in change detection results can be addressed. The calculation of the Dice loss is as follows:

$$L_{dice} = 1 - \frac{2TP}{2TP + FP + FN}, \tag{5}$$

where TP represents the count of correctly predicted positive pixels (true positives), FP represents the count of incorrectly predicted positive pixels (false positives), and FN represents the count of incorrectly predicted negative pixels (false negatives). By considering these values, the Dice loss can effectively evaluate the model in capturing the boundaries and overall similarity of the predicted and true regions in the context of binary classification tasks such as change detection.

The overall loss of MFSFNet is determined by combining $Loss_1$ and $Loss_2$ through summation, and it can be expressed as follows:

$$L_{total} = Loss_1(0.6, 0.4) + Loss_2(0.6, 0.4)$$
$$= 0.6(L_{BCE}(Pre_1, GT) + L_{BCE}(Pre_2, GT)) + 0.4(L_{dice}(Pre_1, GT) + L_{dice}(Pre_2, GT)), \quad (6)$$

where, $Pre_1$ and $Pre_2$ represent the change detection prediction results.

## 4. Experiments

### 4.1. Datasets

To thoroughly evaluate the effectiveness of our method, we conducted experiments on LEVIR-CD and CDD benchmark datasets.

(1)   LEVIR-CD

The LEVIR-CD dataset was introduced by Beihang University. It comprises a large-scale collection of 637 pairs of high-resolution images. The images are of size 1024 × 1024 pixels (0.5 m/pixel) and cover diverse urban and rural areas. This diversity in building types and land cover makes the LEVIR-CD dataset suitable for evaluating the performance of change detection models in complex scenarios. To accommodate memory constraints and sample size, the dataset was divided into non-overlapping patches of size 256 × 256, following the settings in [53]. According to the default dataset partition, the paper used 7120/1024/2048 pairs for training, validation, and testing.

(2)   CDD

The CDD dataset [40] is a popular dataset widely employed for change detection tasks, specifically focusing on seasonal variations in urban landscapes. The dataset comprises real remote sensing images (256 × 256 pixels). One notable characteristic of the CDD dataset is the diverse range of spatial resolutions it offers, spanning from 0.03 to 1 m. Ten thousand pairs were designated for training the model, while 3000 pairs each were allocated for model validation and testing, ensuring an adequate evaluation of the model's performance on unseen data.

### 4.2. Evaluation Metrics

Four commonly used accuracy metrics were employed to assess the experimental results. These metrics are precision, recall, F1 score, and IoU (Intersection over Union). For the binary classification problem of change detection, each pixel can belong to one of two classes. The formulas for calculating the four accuracy metrics are as follows.

$$Precision = \frac{TP}{TP + FP}, \quad (7)$$

$$Recall = \frac{TP}{TP + FN}, \quad (8)$$

$$F1 = 2 \times \frac{Precision \times Recall}{Precision + Recall}, \quad (9)$$

$$IoU = \frac{TP}{FP + TP + FN}. \quad (10)$$

where TP represents the count of correctly predicted positive pixels (true positives), FP represents the count of incorrectly predicted positive pixels (false positives), and FN represents the count of incorrectly predicted negative pixels (false negatives).

### 4.3. Implementation Details

The PyTorch framework was utilized for implementing the proposed model, and all experiments were performed on a GPU, specifically an RTX 3090. The Adam optimizer was utilized with a weight decay of $1 \times 10^{-4}$. To update the learning rate, a cosine annealing

strategy was employed, with a maximum learning rate of $1 \times 10^{-4}$ and a minimum learning rate of $1 \times 10^{-6}$. The learning rate was updated every 20 epochs. During the model training process, several data augmentation techniques were applied to enhance the model's robustness across different scenes. These techniques included random flipping, color augmentation, and multi-scale training. For the experiments conducted on the LEVIR-CD dataset, the model was trained for a total of 60 epochs. Due to the larger sample size of the CDD dataset, the model was trained for a total of 100 epochs. To initialize the feature extraction backbone of the model, pre-trained parameters from ConvNeXt V2-Atto were used, which were obtained through self-supervised training. This initialization helped in leveraging the learned representations from a large-scale dataset and boosted the model's performance.

*4.4. Comparative Methods*

We performed comprehensive experiments comparing it with several SOTA change detection methods. These methods include FC-EF [37], FC-Siam-Diff [37], FC-Siam-Conc [37], STANet [53], BiT [35], and ChangeFormer [64].

- FC-EF [37], FC-Siam-conc [37], and FC-Siam-diff [37] were the first methods to introduce Siamese networks into change detection. EF refers to a fusion technique in which the dual-temporal images are combined or merged at the input stage. Siam-conc represents the concatenation fusion model based on Siamese networks, which combines the dual-temporal features. Siam-diff represents the difference fusion model based on Siamese networks.
- STANet [53] incorporates a self-attention feature fusion module, enabling the model to capture the spatiotemporal dependencies present in various sub-regions of the input images. The self-attention mechanism allows the network to concentrate on important regions and relationships within the images, enhancing its ability to detect changes effectively.
- BiT [35] leverages the Transformer architecture as a change feature fusion network, integrating it with a convolutional neural network (CNN) backbone. This combination enables BiT to capture and model the global semantic information from dual-temporal features. By incorporating Transformers, which excel at capturing long-range dependencies and contextual information, BiT enhances the representation learning process and performance. The CNN backbone complements the Transformer by extracting spatial features from the input images. Together, they form a powerful framework for effectively detecting changes in remote-sensing data.
- ChangeFormer [64] is a change detection method that utilizes a pure Transformer architecture. Unlike traditional methods that combine CNNs and Transformers, ChangeFormer solely relies on Transformers for the entire change detection process. By leveraging the self-attention mechanism, ChangeFormer efficiently captures multi-scale long-range details. The Transformer architecture allows for the modeling of global dependencies and contextual information across the input images, enhancing the overall performance of change detection tasks.

The implemented CD networks above were developed using their publicly available codes.

*4.5. Results Evaluation*

4.5.1. Experimental Results on LEVIR-CD

Table 2 presents the performance of different methods on the LEVIR-CD test set based on the four accuracy metrics. It is worth mentioning that for BiT and ChangeFormer, we directly used the pre-trained model weights provided in their respective papers. In Table 1, the first three fully convolutional neural network methods, FC-EF, FC-Siam-Di, and FC-Siam-Conc, perform poorly. This is mainly because their feature extractors struggle to effectively capture features from complex remote-sensing images. The latter three comparative methods, STANet, BiT, and ChangeFormer, all employ different attention mechanisms

and achieve good performance on the LEVIR-CD dataset. Overall, MFSFNet demonstrated superior performance compared to all six comparative methods across various evaluation metrics. Specifically, it achieved impressive scores of 92.16% for precision, 90.17% for recall, 91.15% for F1 score, and 83.75% for IoU. Notably, our method showcased the most significant improvement in recall. These results highlight the effectiveness and robustness of our approach in accurately detecting changes, particularly in terms of identifying true positive instances.

**Table 2.** The performance of different methods on the LEVIR-CD.

|  | Precision (%) | Recall (%) | F1 (%) | IoU (%) |
|---|---|---|---|---|
| FC-EF | 80.24 | 70.31 | 74.95 | 59.93 |
| FC-Siam-Di | 85.32 | 74.82 | 79.72 | 66.28 |
| FC-Siam-conc | 83.82 | 81.98 | 82.89 | 70.77 |
| STANet | 91.90 | 85.00 | 88.10 | 79.12 |
| BIT | 89.24 | 89.37 | 89.31 | 80.68 |
| ChangeFormer | 92.05 | 88.80 | 90.40 | 82.84 |
| Ours | **92.16** | **90.17** | **91.15** | **83.75** |

Figure 6 visualizes the prediction results of our method and other comparative models on the test set images. As indicated by the accuracy results in Table 2, FC-EF, FC-Siam-Diff, and FC-Siam-Conc performed poorly due to their insufficient feature extraction capabilities. For relatively simple scenes (Figure 6a,e), the three methods STANet, BiT, and ChangeFormer, accurately located the changed buildings, achieving comparable overall performance to our method. However, for relatively complex scenes (Figure 6b–d), our method significantly outperformed all the comparative methods. Specifically, MFSFNet achieved the best results in localizing the changed regions and extracting the boundaries of multi-scale changed regions in complex scenes. The former is attributed to the powerful feature extraction capability of our feature extractor, while the latter relies on the effectiveness of the multi-scale subtraction fusion method.

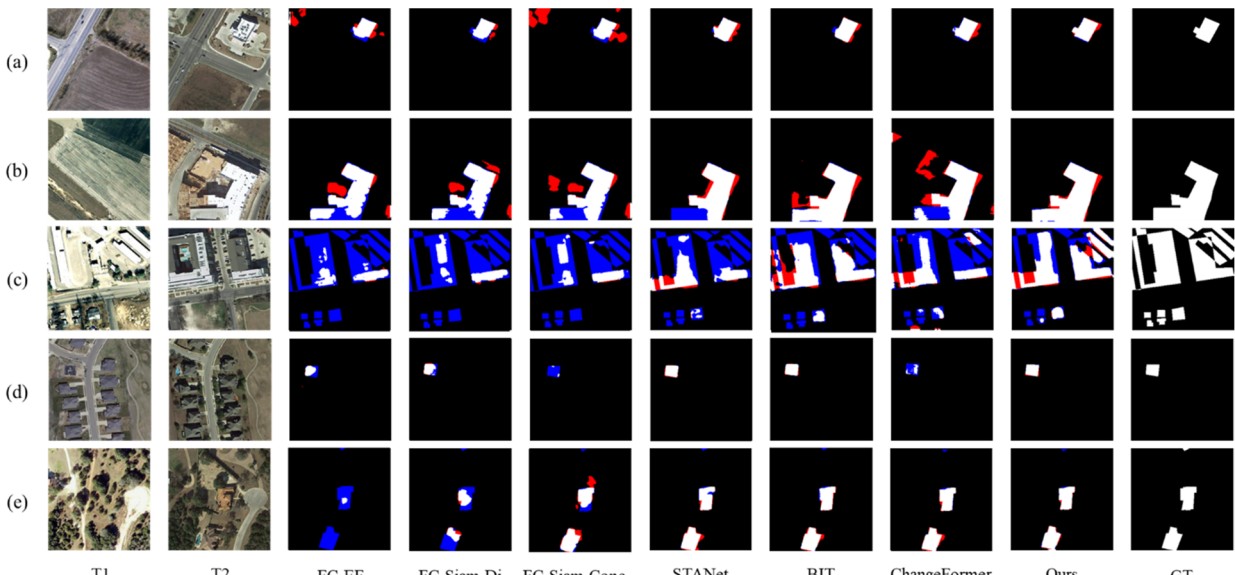

**Figure 6.** The visualizations of the prediction results on the LEVIR-CD dataset. The red pixels in the figure represent false positives, the blue pixels represent false negatives, and the white pixels represent true positives. (**a**–**e**) denote different samples and results in the test set of LEVIR dataset.

4.5.2. Experimental Results on CDD

Table 3 shows that MFSFNet surpasses all the comparative models in terms of recall, F1 score, and IoU, with only a slightly lower precision compared to BiT. Particularly note-

worthy is the high recall achieved by our method, reaching 95.7%, which demonstrates a substantial superiority over other methods and surpasses the ChangeFormer by 2.18%. From an application perspective, recall is more important than precision in change detection tasks, and our effective feature fusion manner helps to lessen the count of missed change regions.

**Table 3.** The performance of different methods on the CDD.

|              | Precision (%) | Recall (%) | F1 (%)    | IoU (%)   |
| ------------ | ------------- | ---------- | --------- | --------- |
| FC-EF        | 66.73         | 54.08      | 59.74     | 42.59     |
| FC-Siam-Di   | 81.51         | 51.68      | 63.25     | 46.25     |
| FC-Siam-conc | 72.60         | 46.58      | 56.75     | 39.62     |
| STANet       | 92.28         | 85.44      | 88.61     | 80.12     |
| BIT          | **96.02**     | 93.26      | 94.61     | 89.78     |
| ChangeFormer | 94.50         | 93.52      | 94.23     | 89.09     |
| Ours         | 95.59         | **95.70**  | **95.64** | **91.65** |

Additionally, the F1 score of the FC-EF, FC-Siam-Di, and FC-Siam-Conc models is only around 60%, much lower than the other four methods, and this performance gap is even more pronounced on the LEVIR-CD dataset. Analyzing the feature extractors of these three methods, we found that their feature extraction capability is weak, making it challenging to accurately locate change regions. We believe that the limited feature extraction capability is the main reason for the poor performance. Furthermore, the images in the CDD dataset exhibit more diverse scenes, such as seasonal variations, which pose additional challenges for feature extraction.

The prediction results of various methods in Figure 7 show our proposed method produced predictions that are closest to the true change regions in the test images. Examining the predicted result images, it is evident that the first three methods struggle to identify change regions in seasonal variations, while the latter four models are able to adapt to some extent to the effects of seasonal changes. In the case of scenes in Figure 7b,c, our model demonstrates more accurate localization of change regions compared to the other models, and the detected change regions are more complete. In the case of the widened road in scene Figure 7e, our proposed method exhibits the best connectivity of change regions, with smoother edges and the closest resemblance to the ground truth.

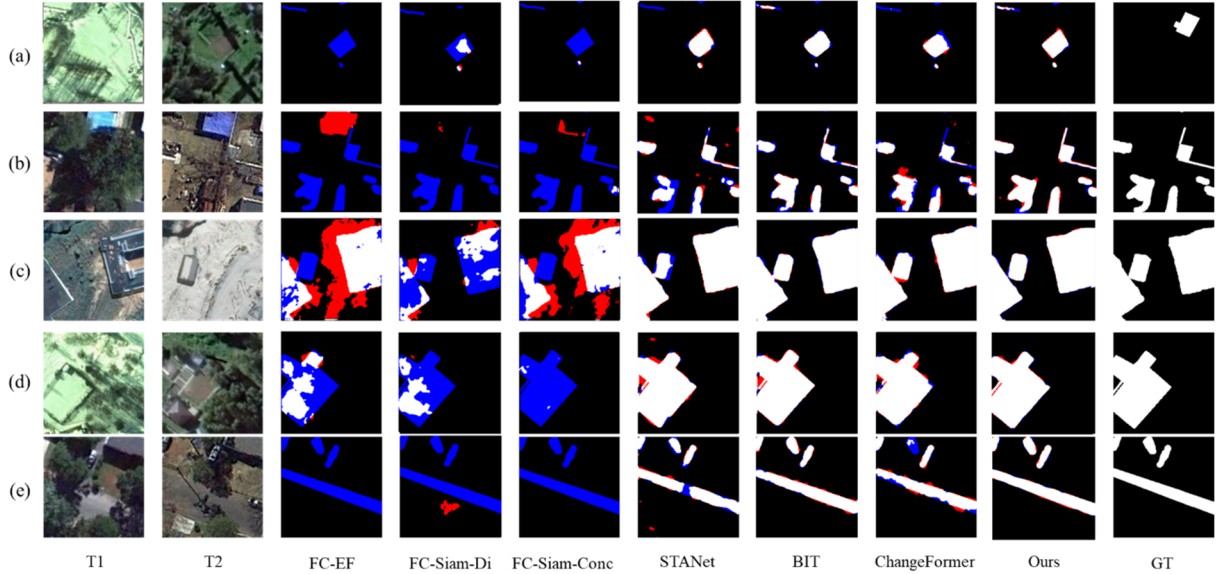

**Figure 7.** The visualizations of the prediction results on the CDD dataset. The red pixels in the figure represent false positives, the blue pixels represent false negatives, and the white pixels represent true positives. (**a**–**e**) denote different samples and results in the test set of CDD dataset.

## 5. Discussion

### 5.1. Ablation Experiments

#### 5.1.1. The Ablation of MFSF and FDS

To conduct a detailed analysis of MFSFNet, ablation experiments were performed on the LEVIR-CD dataset, as summarized in Table 4.

**Table 4.** Ablation experiments on the LEVIR-CD dataset for the MFSF module and FDS module. ($\sqrt{}$ denotes the module is employed).

| Encoder | | MFSF Module | The Stage of Deep Supervision for FDS Module | | | | F1 (%) |
|---|---|---|---|---|---|---|---|
| **Atto** | **Tiny** | | **1** | **2** | **3** | **4** | |
| $\sqrt{}$ | | | | | | $\sqrt{}$ | 88.52 |
| $\sqrt{}$ | | $\sqrt{}$ | | | | $\sqrt{}$ | 89.25 |
| $\sqrt{}$ | | $\sqrt{}$ | | | $\sqrt{}$ | $\sqrt{}$ | **89.51** |
| $\sqrt{}$ | | $\sqrt{}$ | | $\sqrt{}$ | $\sqrt{}$ | $\sqrt{}$ | 89.20 |
| $\sqrt{}$ | | $\sqrt{}$ | $\sqrt{}$ | $\sqrt{}$ | $\sqrt{}$ | $\sqrt{}$ | 89.31 |
| | $\sqrt{}$ | $\sqrt{}$ | | | $\sqrt{}$ | $\sqrt{}$ | **91.15** |

Firstly, we used the lightweight ConvNeXt V2-Atto to get image features. Without using the multi-scale feature subtraction fusion and deep supervision strategy, the F1 score achieved 88.52% for MFSFNet. Next, we successively added the subtraction fusion module and deep supervision at different stages, which resulted in varying degrees of performance improvement. We also conducted a comparative experiment on deep supervision at different stages of the outputs, and the results showed that the model achieved the highest performance of 89.51% when the supervision constraints were applied only in the last two stages. The possible reason for this result is that the feature map sizes of Stage 3 and Stage 4 are closer to the size of the original image compared to Stage 1 and Stage 2. The larger feature maps indicate the presence of more feature information. On the other hand, the feature maps of Stage 1 and Stage 2 contain limited effective information, and directly up-sampling them to the size of the original image leads to poorer performance in change detection results. This, in turn, results in erroneous supervision for the model when all four stages are deep supervised. As a result, the F1 score is lower when all four stages are deep supervised compared to deep supervising only Stage 3 and Stage 4. Finally, we replaced the backbone network with the ConvNeXt V2-Tiny [58], which has a similar scale to ResNet50, and the F1 score further improved to 91.15%, reaching the state-of-the-art performance.

#### 5.1.2. The Effectiveness of MFSF

In this section, a comparison is made between the subtraction fusion strategy and other feature fusion strategies, including product operation, concatenate operation, maximum operation, average operation, and addition operation on the LEVIR-CD dataset. Additionally, we compared the experimental results of using absolute value activation in the subtraction unit with those using ReLU activation to demonstrate the effectiveness of the subtraction fusion strategy in feature fusion.

Table 5 presents the experimental results of change detection on the LEVIR-CD dataset using different feature fusion strategies based on ConvNeXt V2-Atto. The results indicate that our subtraction fusion strategy achieves the highest scores in Precision, F1, and IoU metrics. In terms of the F1, the Product fusion strategy, concatenate fusion strategy, maximum fusion strategy, average fusion strategy, and addition fusion strategy yield similar results, all lower than our subtraction fusion strategy. This is because the change detection task emphasizes the differentiated features between different temporal images, and the subtraction-based feature fusion strategy is more suitable for expressing these differentiated features.

**Table 5.** Comparison of Different Feature Fusion Strategies.

| Fusion Strategy | Precision (%) | Recall (%) | F1 (%) | IoU (%) |
|---|---|---|---|---|
| Product | 90.65 | 87.66 | 89.13 | 80.39 |
| Concatenate | 90.30 | **88.33** | 89.31 | 80.68 |
| Maximum | 90.59 | 87.49 | 89.02 | 80.21 |
| Average | 90.42 | 87.62 | 89.00 | 80.18 |
| Addition | 90.88 | 87.17 | 88.98 | 80.15 |
| Ours | **91.26** | 88.17 | **89.51** | **81.30** |

Table 6 presents the experimental results of change detection on the LEVIR-CD dataset using ReLU and absolute value as activation functions in the Subtraction Unit based on ConvNeXt V2-Atto. The results indicate that using absolute value achieves higher scores than ReLU in Precision, Recall, F1, and IoU metrics. This suggests that using absolute value as the activation function provides a better expression of feature differences, while ReLU discards the negative values in the feature differences, thereby affecting the effectiveness of the subtraction fusion.

**Table 6.** Comparison of Different Activation Function in Subtraction Unit.

| Activation Function | Precision (%) | Recall (%) | F1 (%) | IoU (%) |
|---|---|---|---|---|
| ReLU | 90.80 | 88.12 | 89.40 | 81.16 |
| Absolute Value | **91.26** | **88.17** | **89.51** | **81.30** |

*5.2. Parameter Analysis*

To further analyze the complexity of MFSFNet, we calculated the parameter size and computational complexity of different models in Figure 8. The F1 score of different methods on the LEVIR-CD dataset was represented. Although FC-EF, FC-Siam-Di, and FC-Siam-Conc models have the fewest parameters, their accuracy, and performance are the poorest, making them less suitable for practical scenarios. Therefore, we exclude these three models from the following analysis.

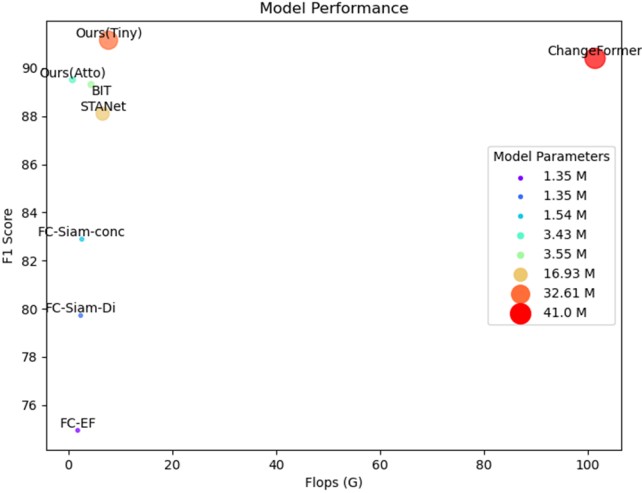

**Figure 8.** Comparison of parameters and FLOPs of different models.

For the other models, our proposed model using ConvNeXt V2-Atto has the lowest FLOPs (floating-point operations per second) while still achieving higher accuracy than STANet and BiT. Regarding ChangeFormer, we assume the use of ChangeFormer V6 [64], which has the maximum parameter size and FLOPs. When we employ ConvNeXt V2-Tiny as the feature extractor in our model, the parameter size of MFSFNet is slightly smaller

compared to ChangeFormer V6, and the number of floating-point operations (FLOPs) is also lower. Despite this, our proposed method achieves the highest F1 score of 91.15% on the LEVIR-CD dataset in this configuration. This demonstrates the efficiency and effectiveness of our approach in achieving superior performance with smaller model size and computational cost compared to the comparative method.

Overall, our proposed model not only achieves superior accuracy compared to other comparative models but also demonstrates better performance in terms of complexity. Our model achieves SOTA results while also having a smaller model size and lower computational cost. This highlights the efficiency and effectiveness of our approach, making it a promising solution for change detection tasks.

## 6. Conclusions

In this paper, we propose MFSFNet for change detection in remote sensing images. The key contributions of our approach lie in its twin lightweight encoder-decoder structure and the incorporation of the multi-scale feature subtraction fusion and feature deep supervision modules. The MFSFNet architecture is designed to accurately extract change features. The multi-scale feature subtraction fusion module enhances the representation of change features by emphasizing the differences at various scales while reducing the influence of irrelevant pseudo-change features. This module plays a crucial role in capturing meaningful change regions. To further improve the training performance and enable effective feature learning, we introduce the feature deep supervision module. This module provides additional supervision to the change features at different scales in the decoder, promoting the learning of discriminative representations. By leveraging multi-scale deep supervision, our network effectively captures hierarchical information inherent in the change detection task. With the Dice loss along, our method mitigates the class imbalance problem and reduces boundary fuzziness in the change detection results. The experimental results demonstrate the superior performance of our MFSFNet compared to existing state-of-the-art methods. Our approach achieves a better balance between complexity and performance, indicating its robustness and effectiveness in change detection tasks. In future work, we will focus on making network models and change detection solutions more lightweight, applicable to small sample sizes, and adaptable to weak supervision. This will contribute to the rapid and accurate extraction of surface change information, reducing excessive reliance on change samples and thus improving the applicability of this method in regular surface monitoring.

**Author Contributions:** Conceptualization, Z.H. and H.Y.; methodology, Z.H.; software, Z.H.; validation, Z.H.; formal analysis, Z.H.; investigation, Z.H.; resources, Z.H.; data curation, Z.H.; writing—original draft preparation, Z.H.; writing—review and editing, Z.H.; visualization, Z.H.; supervision, H.Y.; project administration, H.Y.; funding acquisition, H.Y. All authors have read and agreed to the published version of the manuscript.

**Funding:** This research received no external funding.

**Data Availability Statement:** Not applicable.

**Conflicts of Interest:** All authors declare no conflict of interest.

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
