# Peer review of "MFSFNet: Multi-Scale Feature Subtraction Fusion Network for Remote Sensing Image Change Detection"

_remotesensing, doi:10.3390/rs15153740_

Round 1
Reviewer 1 Report
How to fuse the information of multi-temporal image is the main issue in change detection. The main contribution of this paper is the multi-scale feature subtraction fusion (MSSF) module. Therefore, the reviewer mainly focuses on MSSF. Overall, this paper has shown promising performance on two benchmark datasets. However, the reviewer suggests the author making more analysis on MSSF.
1. In Eq.(1), the subtraction unit is applied to features between two scales. The authors used the absolute value operation. In the view of this reviewer, absolute value operation actually acts as the role of activation function. The authors did not explain why absolute value operation should be exploited here, and there is no ablation experienments on this activation function. The authors must conduct an extra ablation experiment to explore whether other activation functions provide better performance or not.
2. In addition, please refer to table 1 in [R1], where Product, Concatenation, Max, Mean and Addition are employed to fuse information. The authors must carry out the similar ablation experiment on fusion strategy to prove subtraction is the better operation for information fusion.
[R1] Ram Prabhakar, K. and Sai Srikar, V and Venkatesh Babu, R.DeepFuse: A Deep Unsupervised Approach for Exposure Fusion With Extreme Exposure Image Pairs. ICCV 2017.
3. Table 4 is suggested to be shown as a figure, which would be a more friendly way to display the results.
In the second round, there are still typos and editorial errors.
Author Response
Response to Reviewer 1 Comments:
How to fuse the information of multi-temporal image is the main issue in change detection. The main contribution of this paper is the multi-scale feature subtraction fusion (MSSF) module. Therefore, the reviewer mainly focuses on MSSF. Overall, this paper has shown promising performance on two benchmark datasets. However, the reviewer suggests the author making more analysis on MSSF.
Response: We are very grateful to you for your careful review of our study.
- In Eq.(1), the subtraction unit is applied to features between two scales. The authors used the absolute value operation. In the view of this reviewer, absolute value operation actually acts as the role of activation function. The authors did not explain why absolute value operation should be exploited here, and there are no ablation experiments on this activation function. The authors must conduct an extra ablation experiment to explore whether other activation functions provide better performance or not.
Response: Thank you very much for your suggestion. We included an explanation of using the absolute value activation function in Section 3.3 of the revised manuscript. In Section 5.1.2, we discussed the results of ablation experiments on the absolute value activation function.
- In addition, please refer to table 1 in [R1], where Product, Concatenation, Max, Mean and Addition are employed to fuse information. The authors must carry out the similar ablation experiment on fusion strategy to prove subtraction is the better operation for information fusion.
[R1] Ram Prabhakar, K. and Sai Srikar, V and Venkatesh Babu, R.DeepFuse: A Deep Unsupervised Approach for Exposure Fusion With Extreme Exposure Image Pairs. ICCV 2017.
Response: Thank you very much for your suggestion. We conducted ablation experiments on feature fusion strategies, following the reference [R1], and the results demonstrated the effectiveness of the subtraction fusion strategy. You can find the relevant content in Section 5.1.2 of the revised manuscript.
- Table 4 is suggested to be shown as a figure, which would be a more friendly way to display the results.
Response: Thank you very much for your suggestion. We have transformed Table 4 into a bubble chart, where the horizontal axis represents FLOPs, the vertical axis represents F1, the bubble size represents the number of parameters, and the bubble color represents different models. As shown in Figure 8 of the newly submitted manuscript.
Reviewer 2 Report
Article ID:remotesensing-2475539
In this paper, the authors contributed the following work
l It proposed the MFSFNet for high-resolution remote sensing image change detection. T
l It utilized a lightweight feature extraction network and introduce a novel deep supervision strategy in the change decoder, which enhances the training performance of the network
l It demonstrated that MFSFNet surpasses these recent state-of-the-art models, achieving the highest F1 scores of 91.15% and 95.64% on the LEVIR-CD and CDD datasets, respectively.
Therefore, it is interesting and attractive. However, it should be minor revised to enhance the quality, as follows:
1) Literature review is not upto the mark requreid a summary table which will provide with comparison analysis of other approaches
2) There should be one flow chart for overall work methodology.
3) Figure 1 and 4 should be re-presented. Moreover, all the parameters should be explain clearly .
4) Eq,7 to 9 are not derived properly, Pl explain it in details .
5) Pl explain the future research direction in the conclusion section
Author Response
Response to Reviewer 2 Comments:
In this paper, the authors contributed the following work
l It proposed the MFSFNet for high-resolution remote sensing image change detection. T
l It utilized a lightweight feature extraction network and introduce a novel deep supervision strategy in the change decoder, which enhances the training performance of the network
l It demonstrated that MFSFNet surpasses these recent state-of-the-art models, achieving the highest F1 scores of 91.15% and 95.64% on the LEVIR-CD and CDD datasets, respectively.
Response: We are very grateful to you for your careful review of our study.
Therefore, it is interesting and attractive. However, it should be minor revised to enhance the quality, as follows:
1) Literature review is not upto the mark requreid a summary table which will provide with comparison analysis of other approaches
Response: Thank you very much for your suggestion. We have summarized the literature review in Table 1, which compares the encoder architectures, feature fusion strategies, and publication years of different change detection methods.
2) There should be one flow chart for overall work methodology.
Response: Thank you very much for pointing out this problem for us. In Figure 1 of the newly submitted manuscript, we have presented the flowchart of our proposed method.
3) Figure 1 and 4 should be re-presented. Moreover, all the parameters should be explain clearly .
Response: Thank you very much for pointing out this problem for us. In the newly submitted manuscript, we have provided explanations for the original Figure 1 and Figure 4 in Figure 2 and Figure 5, respectively.
4) Eq,7 to 9 are not derived properly, Pl explain it in details .
Response: Thank you very much for pointing out this problem for us. In the newly submitted manuscript, we have provided explanations for the parameters used in our method.
5) Pl explain the future research direction in the conclusion section
Response: Thank you very much for pointing out this problem for us. In the conclusion section of the newly submitted manuscript, we have added a discussion on future directions for further research.
Reviewer 3 Report
This paper presents a detailed experiment of comparison of deep learning arquitectures for image change detection. It is well structured and clear. I am enclosing your PDF manuscript with minor comments.
There is a general issue I would like to comment: most of your references are of Asian authors; even those of the Introduction and Related Works sections, that are more general. I would suggest that you include other European and American authors working in this field in order to aknowledge, respect, and give credit to their job.

Author Response
Response to Reviewer 3 Comments:
This paper presents a detailed experiment of comparison of deep learning arquitectures for image change detection. It is well structured and clear. I am enclosing your PDF manuscript with minor comments.
Response: We are very grateful to you for your careful review of our study. We have carefully reviewed the comments in the PDF and made revisions to the manuscript accordingly.
There is a general issue I would like to comment: most of your references are of Asian authors; even those of the Introduction and Related Works sections, that are more general. I would suggest that you include other European and American authors working in this field in order to aknowledge, respect, and give credit to their job.
Response: Thank you very much for your suggestion. In the newly submitted manuscript, we have added references from European and American authors in the introduction and related work sections.
Reviewer 4 Report
The paper introduces MFSF-Net, a Multi-Scale Feature Subtraction Fusion Network, to improve change detection in remote sensing images. The proposed approach effectively enhances change features and reduces redundant pseudo-change features, addressing existing challenges in the field. The incorporation of the FDS module for additional supervision and the use of the Dice loss strategy to handle imbalanced samples demonstrate the authors' thorough approach. The paper's experimental results showcase its superiority over state-of-the-art algorithms.
However, other change analysis approaches in the literature should also be mentioned. For example, change analysis studies on temporal images use the 3D CNN model. These need to be mentioned.
Author Response
Response to Reviewer 4 Comments:
The paper introduces MFSF-Net, a Multi-Scale Feature Subtraction Fusion Network, to improve change detection in remote sensing images. The proposed approach effectively enhances change features and reduces redundant pseudo-change features, addressing existing challenges in the field. The incorporation of the FDS module for additional supervision and the use of the Dice loss strategy to handle imbalanced samples demonstrate the authors' thorough approach. The paper's experimental results showcase its superiority over state-of-the-art algorithms.
Response: We are very grateful to you for your careful review of our study
However, other change analysis approaches in the literature should also be mentioned. For example, change analysis studies on temporal images use the 3D CNN model. These need to be mentioned.
Response: Thank you very much for your suggestion. In the related work section, we have added a review of 3D CNN-based methods for change detection.
Reviewer 5 Report
The paper propose a new framework named MFSF-Net for CD in HR images. The network includes two modules and four stages. The framework was proposed to overcome some challenges. However this work is not very clear. The whole process was more like a strategy of several general processes, although it can achieve better results than recently methods. The novelty of the manuscripts should be strengthened. The core algorithm needed to be detailed. Here (just for indication) some ways to improve the current version of the paper:
1. You mentioned that the existing change feature fusion methods often introduce redundant information, meanwhile, the structure of network is complex. However, the proposed framework didn’t fix the problems according to these issues. Please clarify how to make the algorithm faster and reduce the parameters.
2. In section 3.4, why choose stage 2 is chosen and fused with SF2 and generate loss2. How can we produce the Multi-scale feature deep supervision module? I think the paragraph needs to be detailed.
3. In the experiment, several traditional methods are implemented. In the Tab.1, why the first four methods only get the poor results? The highest precision is only 91.9%, which is not convinced. In my opinion, theses methods can get more reliable results in the other literature.
4. In the line426, you mentioned that the highest F1 score did not exceeding 70%. How can you get this conclusion. In the Tab.1, the FC-EF, the worst method can get 74.95%.
5. I suggested that the Fig.5 and Fig.6 both needs to label the changes with different colors. In the current version, the tiny changes cannot be captured.
6. In the ablation experiment, the different modules were implemented to test the model. I cannot understand why the highest performance of 89.51% is achieved when last two stages were applied.
The quality of English language can be accepted.
Author Response
Response to Reviewer 5 Comments:
The paper propose a new framework named MFSF-Net for CD in HR images. The network includes two modules and four stages. The framework was proposed to overcome some challenges. However this work is not very clear. The whole process was more like a strategy of several general processes, although it can achieve better results than recently methods. The novelty of the manuscripts should be strengthened. The core algorithm needed to be detailed. Here (just for indication) some ways to improve the current version of the paper:
Response: We are very grateful to you for your careful review of our study
1.You mentioned that the existing change feature fusion methods often introduce redundant information, meanwhile, the structure of network is complex. However, the proposed framework didn’t fix the problems according to these issues. Please clarify how to make the algorithm faster and reduce the parameters.
Response: Thank you very much for your comments. Indeed, in Section 5.2, we analyzed the parameter count and FLOPs of different change detection models. Our proposed model not only achieves higher accuracy compared to other comparison models but also exhibits better performance in terms of complexity. Our model achieves state-of-the-art results while having a smaller model size and lower computational costs. This highlights the efficiency and effectiveness of our approach, making it a promising solution for change detection tasks.
2.In section 3.4, why choose stage 2 is chosen and fused with SF2 and generate loss2. How can we produce the Multi-scale feature deep supervision module? I think the paragraph needs to be detailed.
Response: Thank you very much for your comments. In the newly submitted manuscript, we have added a section, Section 3.4, that discusses the use of the feature deep supervision module and the rationale behind applying deep supervision in stage 2.
- In the experiment, several traditional methods are implemented. In the Tab.1, why the first four methods only get the poor results? The highest precision is only 91.9%, which is not convinced. In my opinion, theses methods can get more reliable results in the other literature
Response: Thank you very much for your comments. Firstly, in all the comparative experiments, we directly used the open-source code or implemented the methods based on the architectures and parameters provided in the papers without any modifications. Therefore, the experimental results are trustworthy and credible.
Secondly, regarding FC-EF, FC-Siam-Di, and FC-Siam-conc, they were proposed in 2018. In the original paper, the authors used a convolutional architecture with a 16-layer backbone[R1]. However, a simplistic backbone architecture poses significant challenges in handling the current change detection datasets. Moreover, previous studies have shown that increasing the depth of the backbone used in the FC series significantly improves accuracy[R2]. Other literature reporting higher accuracy may have deepened the backbone architecture, whereas we have used the original architecture as described in the original papers.
Thirdly, the visual comparison results in the experimental visualization clearly show that the results of FC-EF, FC-Siam-Di, and FC-Siam-conc are significantly worse than other methods, which is consistent with the lower evaluation scores.
Lastly, we investigated the literature that utilized FC-EF, FC-Siam-Di, and FC-Siam-conc methods on the LEVIR-CD dataset. We found that there were cases where different papers reported exactly the same values for the assessment metrics [R3, R4], which we consider highly unlikely. Even when using the same model trained and tested on the same dataset, it is impossible to achieve identical results due to factors such as device variations, optimization processes, and random noise. Therefore, we consider the excessively high results obtained in other literature to be less reliable.
[R1]Daudt, R. C., Le Saux, B., & Boulch, A. (2018, October). Fully convolutional siamese networks for change detection. In 2018 25th IEEE International Conference on Image Processing (ICIP) (pp. 4063-4067). IEEE.
[R2] Fang, S., Li, K., Shao, J., & Li, Z. (2021). SNUNet-CD: A densely connected Siamese network for change detection of VHR images. IEEE Geoscience and Remote Sensing Letters, 19, 1-5.
[R3] Bandara, W. G. C., & Patel, V. M. (2022, July). A transformer-based siamese network for change detection. In IGARSS 2022-2022 IEEE International Geoscience and Remote Sensing Symposium (pp. 207-210). IEEE.
[R4] Chen, H., Qi, Z., & Shi, Z. (2021). Remote sensing image change detection with transformers. IEEE Transactions on Geoscience and Remote Sensing, 60, 1-14.
- In the line426, you mentioned that the highest F1 score did not exceeding 70%. How can you get this conclusion. In the Tab.1, the FC-EF, the worst method can get 74.95%.
Response: Thank you very much for pointing out this problem for us. We apologize for the writing error, and we have removed this incorrect statement in the newly submitted manuscript.
- I suggested that the Fig.5 and Fig.6 both needs to label the changes with different colors. In the current version, the tiny changes cannot be captured.
Response: Thank you very much for your suggestion. We have modified the representation of these two figures by using different color schemes to depict the results of change detection. The revised images can be found in Figures 6 and 7 of the new version of the manuscript.
- In the ablation experiment, the different modules were implemented to test the model. I cannot understand why the highest performance of 89.51% is achieved when last two stages were applied.
Response: Thank you very much for your comments. In the newly submitted manuscript, we have added a discussion in Section 5.1 about the reasons why supervising the last two stages leads to optimal results.
Round 2
Reviewer 5 Report
I have not found any major items to correct in this manuscript.
English launguage is fine.